# Panton–Valentine leucocidin is the key determinant of *Staphylococcus aureus* pyomyositis in a bacterial GWAS

Bernadette C Young[1,2], Sarah G Earle[1], Sona Soeng[3], Poda Sar[3], Varun Kumar[4], Songly Hor[3], Vuthy Sar[3], Rachel Bousfield[5], Nicholas D Sanderson[1], Leanne Barker[1], Nicole Stoesser[1,6], Katherine RW Emary[2], Christopher M Parry[7,8], Emma K Nickerson[5], Paul Turner[3,9], Rory Bowden[10], Derrick W Crook[1,2,6], David H Wyllie[1,6,11], Nicholas PJ Day[9], Daniel J Wilson[1,10,12], Catrin E Moore[9,13]*

[1]Nuffield Department of Medicine, Experimental Medicine Division, University of Oxford, John Radcliffe Hospital, Oxford, United Kingdom; [2]NIHR Oxford Biomedical Research Centre, Infection Theme, Oxford University Hospitals NHS Foundation Trust, John Radcliffe Hospital, Oxford, United Kingdom; [3]Cambodia Oxford Medical Research Unit, Angkor Hospital for Children, Siem Reap, Cambodia; [4]Department of Pediatrics, East Tennessee State University Quillen College of Medicine, Johnson City, United States; [5]Department of Infectious Diseases, Cambridge University Hospitals NHS Foundation Trust, Cambridge, United Kingdom; [6]Public Health England Academic Collaborating Centre, John Radcliffe Hospital, Oxford, United Kingdom; [7]Clinical Sciences, Liverpool School of Tropical Medicine, Liverpool, United Kingdom; [8]School of Tropical Medicine and Global Health, Nagasaki University, Nagasaki, Japan; [9]Centre for Tropical Medicine and Global Health, Nuffield Department of Medicine, University of Oxford, Oxford, United Kingdom; [10]Wellcome Centre for Human Genetics, University of Oxford, Oxford, United Kingdom; [11]The Jenner Institute Laboratories, University of Oxford, Oxford, United Kingdom; [12]Institute for Emerging Infections, Oxford Martin School, University of Oxford, Oxford, United Kingdom; [13]Mahidol-Oxford Tropical Medicine Research Unit, Faculty of Tropical Medicine, Mahidol University, Bangkok, Thailand

*For correspondence:
catrin.moore@ndm.ox.ac.uk

Competing interests: The authors declare that no competing interests exist.

**Abstract** Pyomyositis is a severe bacterial infection of skeletal muscle, commonly affecting children in tropical regions, predominantly caused by *Staphylococcus aureus*. To understand the contribution of bacterial genomic factors to pyomyositis, we conducted a genome-wide association study of *S. aureus* cultured from 101 children with pyomyositis and 417 children with asymptomatic nasal carriage attending the Angkor Hospital for Children, Cambodia. We found a strong relationship between bacterial genetic variation and pyomyositis, with estimated heritability 63.8% (95% CI 49.2–78.4%). The presence of the Panton–Valentine leucocidin (PVL) locus increased the odds of pyomyositis 130-fold ($p=10^{-17.9}$). The signal of association mapped both to the PVL-coding sequence and to the sequence immediately upstream. Together these regions explained over 99.9% of heritability (95% CI 93.5–100%). Our results establish staphylococcal pyomyositis, like tetanus and diphtheria, as critically dependent on a single toxin and demonstrate the potential for association studies to identify specific bacterial genes promoting severe human disease.
DOI: https://doi.org/10.7554/eLife.42486.001

**eLife digest** Certain bacteria that normally live on the skin or in the nose without causing problems can sometimes lead to diseases elsewhere in the body. For example, the bacterium *Staphylococcus aureus* can cause blood infections or a severe and painful infection of the muscle called pyomyositis, which is very common in children who live in the tropics.

Scientists believe that pyomyositis happens when *S. aureus* bacteria in the blood stream infect the muscles. Some strains of this bacteria are more likely to cause such infections, but why is unclear. One potential cause is a toxin produced by some *S. aureus* bacteria called Panton-Valentine leucocidin (PVL). So far, studies looking at whether PVL-producing bacteria are more likely to cause pyomyositis have had conflicting results.

Now, Young et al. show that the gene for PVL is always present in *S. aureus* strains that cause pyomyosistis in Cambodian children, but is rarely found in *S. aureus* taken from the noses of their healthy counterparts. In the experiments, bacteria were collected from 101 children with pyomyositis and from the noses of 417 healthy children at the Angkor Hospital for Children in Cambodia over a 5-year period. The DNA in these bacteria were compared using very sensitive genetic techniques. The comparisons showed having the gene for PVL increased the odds of having pyomyositis 130-fold, showing that this one toxin likely accounts for much of the risk of developing this disease.

If more studies confirm the link between PVL and pyomyositis, developing vaccines that block the gene for PVL might be one way to protect children in the tropics from developing this infection. Treating children with pyomyositis with antibiotics that reduce the production of the PVL toxin may also be helpful.

DOI: https://doi.org/10.7554/eLife.42486.002

## Introduction

Microbial genome sequencing and bacterial genome-wide association studies (GWAS) present new opportunities to discover bacterial genes involved in the pathogenesis of serious infections (*Sheppard et al., 2013*; *Chewapreecha et al., 2014*; *Earle et al., 2016*; *Lees et al., 2016*; *Falush, 2016*; *Power et al., 2017*). Pyomyositis is a severe infection of skeletal muscle most commonly seen in children in the tropics (*Chauhan et al., 2004*; *Verma, 2016*; *Bickels et al., 2002*). In up to 90% of cases, it is caused by a single bacterial pathogen, *Staphylococcus aureus* (*S. aureus*) (*Chauhan et al., 2004*; *Verma, 2016*; *Bickels et al., 2002*; *Moriarty et al., 2015*). Unlike infections of the skin and superficial soft tissues, the skin and subcutaneous tissues are not usually involved in pyomyositis, by contrast to intense inflammation in the infected muscles (*Chauhan et al., 2004*; *Verma, 2016*). Pyomyositis is thought to arise from haematogenous seeding of bacteria from blood to muscle (*Verma, 2016*). There is evidence that some *S. aureus* strains have heightened propensity to cause pyomyositis – the incidence in the USA doubled during an epidemic of community-associated methicillin resistant *S. aureus* (CA-MRSA) (*Pannaraj et al., 2006*) – but molecular genetic investigation of *S. aureus* from pyomyositis has been limited (*Borges et al., 2012*).

Panton–Valentine leucocidin (PVL), a well-known staphylococcal toxin causing purulent skin infections and found in epidemics caused by CA-MRSA, has been implicated in pyomyositis, pneumonia and other *S. aureus* disease manifestations, but its role in these invasive infections is disputed (*Shallcross et al., 2013*; *Vandenesch et al., 2003*; *Labandeira-Rey et al., 2007*; *Villaruz et al., 2009*). PVL is a bipartite pore-forming toxin comprising the co-expressed LukF-PV and LukS-PV proteins (*Löffler et al., 2010*; *Boakes et al., 2011*). The coding sequence for PVL, *lukSF-PV*, is usually carried on bacteriophages, (*Shallcross et al., 2013*; *Löffler et al., 2010*), which facilitate *lukSF-PV* exchange between lineages (*McCarthy et al., 2012*). The mechanism of PVL toxicity has been shown to involve cell lysis in human myeloid cells, particularly neutrophils, by insertion into the cellular membrane, (*Niemann et al., 2018*) leading the tissue to release inflammatory neutrophil products (*Spaan et al., 2013*). Neutrophil lysis is mediated by PVL binding to target complement receptors C5aR; in binding, PVL has both toxic and immunomodulatory effects, as it also inhibits C5a mediated immune activation (*Sina et al., 2013*).

Although small case series testing for candidate genes have reported a high prevalence of PVL among pyomyositis-causing *S. aureus*, (*Pannaraj et al., 2006*; *Sina et al., 2013*; *García et al., 2013*)

a detailed meta-analysis found no evidence for an increased rate of musculoskeletal infection (or other invasive disease) in PVL-positive bacteria versus controls (*Shallcross et al., 2013*). These conflicting results may reflect insufficiently powered studies, and some case series lack comparative control strains (*Sina et al., 2013*). A further problem with the use of candidate gene studies in studying pathogenesis is that they may miss important variation elsewhere in the genome. One study reporting a critical role for PVL in the causation of severe pneumonia (*Labandeira-Rey et al., 2007*) was later found to have overlooked mutations in key regulatory genes, capable of producing the virulence that had been attributed to PVL by the original study (*Villaruz et al., 2009*). Thus, while some evidence suggests an association between pyomyositis and PVL, there remains significant uncertainty regarding the bacterial genetic predisposition of *S. aureus* to pyomyositis, and whether PVL is an important virulence factor, or merely an epiphenomenon, carried by bacteria alongside unidentified genetic determinants (*Otto, 2011*; *Day, 2013*).

GWAS offer a means to screen entire bacterial genomes to discover genes and genetic variants associated with disease risk. They are particularly appealing because they enable the investigation of traits not readily studied in the laboratory, and do not require the nomination of specific candidate genes (*Falush, 2016*). Proof-of-principle GWAS in bacteria have demonstrated the successful rediscovery of known antimicrobial resistance (AMR) determinants (*Chewapreecha et al., 2014*; *Earle et al., 2016*; *Lees et al., 2016*). However, AMR is under extraordinarily intense selection in bacteria. More subtle traits, including host specificity (*Sheppard et al., 2013*) and the duration of pneumococcal carriage, (*Lees et al., 2017*) have also been demonstrated using GWAS. Promising results for GWAS in human infecting bacteria include identifying possible loci for invasive infection with *Streptococcus pyogenes* (*Lees et al., 2016*) and *Staphylococcus epidermidis* (*Méric et al., 2018*),and the identification of virulence-associated genes corresponding with regional differences in disease manifestations of melioidosis (*Chewapreecha et al., 2017*). Within species, lineage-specific variants have been shown to predict mortality following *S. aureus* bacteraemia (*Recker et al., 2017*). These studies support the potential GWAS has to precisely pinpoint genes and genetic variants underlying the propensity to cause specific human infections, making it a promising tool to investigate the possible contribution of bacterial genomic variation to pyomyositis.

## Results

To understand the bacterial genetic basis of pyomyositis, we sampled and whole-genome sequenced *S. aureus* from 101 pyomyositis infections and 417 asymptomatic nasal carriage episodes in 518 children attending Angkor Hospital for Children in Siem Reap, Cambodia between 2008 and 2012 (*Supplementary file 1*). As expected, we observed representatives of multiple globally common lineages in Cambodia, together with some globally less common lineages at high frequency, in particular clonal complex (CC) 121, identified by multi-locus sequence typing (MLST). There were no major changes in lineage frequency over time (*Figure 1—figure supplement 1*).

In our study, some *S. aureus* lineages were strongly overrepresented among cases of pyomyositis compared with asymptomatic, nasally-carried controls over the same time period. Notably, 86/101 (85%) of pyomyositis cases were caused by CC-121 bacteria, whereas no pyomyositis cases were caused by the next two most commonly carried lineages, sequence type (ST)−834 and CC-45 (*Figure 1*). We estimated the overall heritability of case/control status to be 63.8% (95% CI 49.2–78.4%) in the sample, reflecting the strong relationship between bacterial genetic variation and case/control status. We used *bugwas* (*Power et al., 2017*) to decompose this heritability into the principal components (PCs) of bacterial genetic variation. PC 1, which distinguished CC-121 (the most common pyomyositis lineage) from ST-834 (which was only found in carriage), showed the strongest association with case/control status ($p=10^{-29.6}$, Wald test). The strongest association was with PC 20, which differentiated a sub-lineage of CC-121 within which no cases were seen ($p=10^{-13.9}$), and PC 2, which distinguished CC-45 from the rest of the species ($p=10^{-4.9}$).

We conducted a GWAS to identify bacterial genetic variants associated with pyomyositis, controlling for differences in pyomyositis prevalence between *S. aureus* lineages. We used a kmer-based approach (*Sheppard et al., 2013*) in which every variably present 31 bp DNA sequence observed among the 518 genomes was tested for association with pyomyositis *versus* asymptomatic nasal carriage, controlling for population structure using GEMMA (*Zhou and Stephens, 2012*). These kmers captured bacterial genetic variation including single nucleotide polymorphisms (SNPs), insertions or

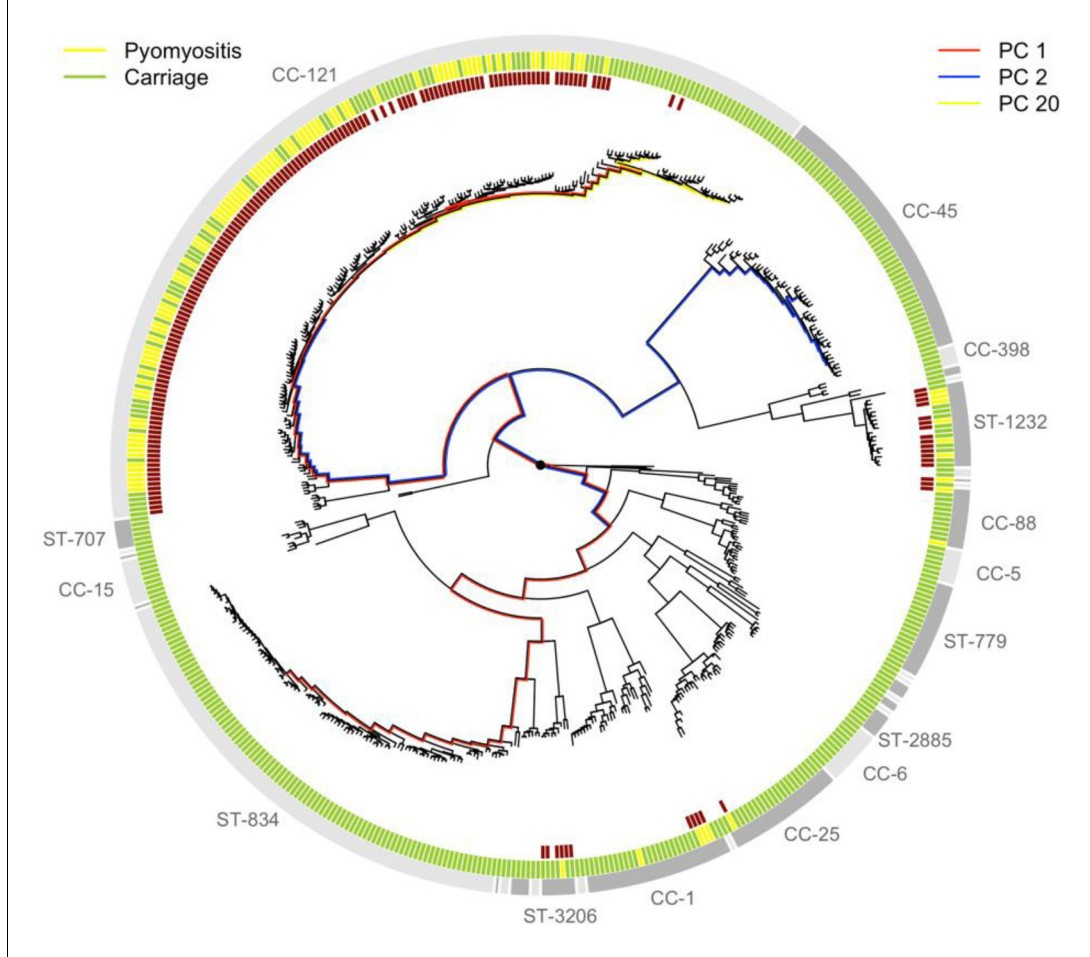

**Figure 1.** Phylogeny of *S. aureus* cultured from children in Cambodia shows strong strain-to-strain variation in pyomyositis prevalence. The phylogeny was estimated by maximum likelihood from SNPs mapping to the USA300 FPR3757 reference genome. Multi-locus sequence type (ST) or clonal complex (CC) groups are shown (outer grey ring). Case/control status is marked in the middle ring: pyomyositis (gold, n = 101) or nasal carriage (green, n = 417). Branches of the phylogeny that correspond to the three principal components (PCs) significantly associated with case/control status (PCs 1, 2 and 20) are marked in red, blue and yellow, respectively. Branch lengths are square root transformed to aid visualization. The presence of the kmers most strongly associated with pyomyositis is indicated by red blocks in the inner ring.

DOI: https://doi.org/10.7554/eLife.42486.003

The following figure supplement is available for figure 1:

**Figure supplement 1.** Sampling frequencies of the major strains were stable over time.

DOI: https://doi.org/10.7554/eLife.42486.004

deletions (indels), and presence or absence of entire accessory genes. We found 10.7 million unique kmers variably present across the bacterial genomes. In total, 9175 kmers were significantly associated with case/control status after correction for multiple testing ($10^{-6.8} \leq p \leq 10^{-21.4}$; *Figure 2A*). When mapped to the de novo assembly of a CC121 isolate from pyomyositis (PYO2134), the vast majority of these kmers (9,074/9,175; 98.9%) localised to a 45.7 kb region spanning an integrated prophage with 95% nucleotide identity to φSLT (*Figure 2B*). Most (9,173/9,175; 99.98%) significant kmers were found at an increased frequently in pyomyositis, with odds ratios (OR) ranging from 2.7 to 139.8, indicating that the presence of each was associated with increased risk of disease. The presence of bacteriophage φSLT was thus strongly associated with pyomyositis.

We were able to localise the most statistically significant signal of association to kmers that mapped within φSLT to the *lukS-PV* and *lukF-PV* cargo genes. These genes encode the subunits of PVL, which multimerise into a pore-forming toxin capable of rapidly lysing the membranes of human neutrophils (*Löffler et al., 2010*; *Otto, 2011*). 1630 kmers tagging the presence of the *lukSF-PV*

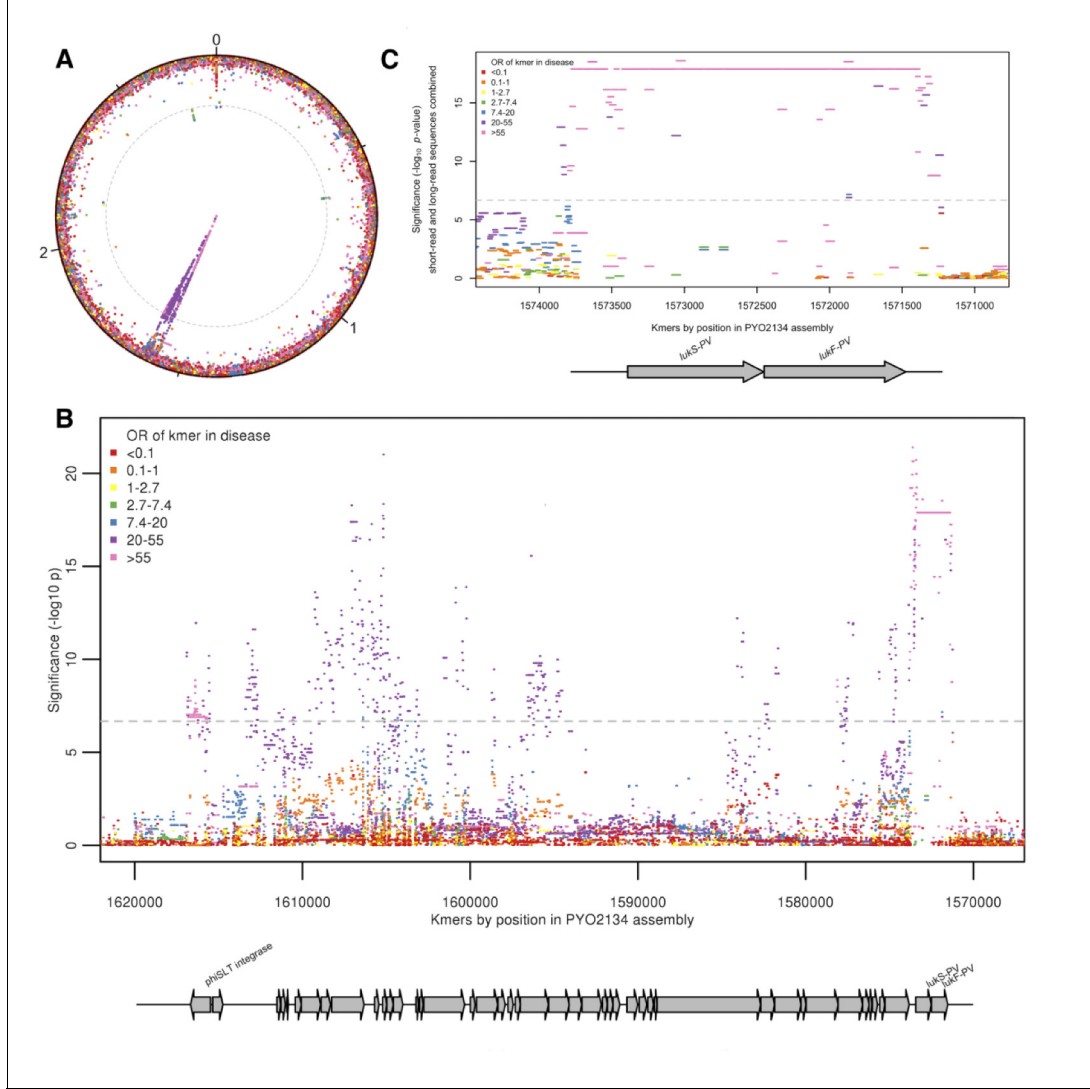

**Figure 2.** Kmers associated with pyomyositis.  (A) All kmers (n = 10,744,013) were mapped to the genome assembly of one CC121 pyomyositis bacterium (PYO2134). Each point represents a kmer, plotted by the mapped location and the significance of the association with disease (-log₁₀ *p* value). Kmers are coloured by the odds ratio (OR) of kmer presence for disease risk. A Bonferroni-adjusted threshold for significance is plotted in grey. (B) The region between 1.57–1.62 MB in greater detail. Grey arrows depict coding sequences, determined by homology to USA300 FPR3757. (C) Associations for kmers mapping to region 1,571–1574 kB is plotted. Kmer presence determined from hybrid assembly using short and long-read data. Grey arrows depict coding sequences, determined by homology to USA300 FPR3757.

DOI: https://doi.org/10.7554/eLife.42486.005

The following figure supplements are available for figure 2:

**Figure supplement 1.** Alignments of reference genome PYO2134 assembly (R) with 37 de novo assemblies of Illumina short-read sequencing (C) which feature either ambiguities (Ns) or contig boundaries in the region 389 bp upstream of PVL coding sequence.

DOI: https://doi.org/10.7554/eLife.42486.006

**Figure supplement 2.** Alignments of reference genome PYO2134 assembly (R) with 37 de novo assemblies of Illumina short-read sequencing (C) which feature either ambiguities (Ns) or contig boundaries in the region 389 bp upstream of PVL coding sequence.

DOI: https://doi.org/10.7554/eLife.42486.007

**Figure supplement 3.** Presence of PVL and potential PVL carrying phages across the population.

DOI: https://doi.org/10.7554/eLife.42486.008

**Figure supplement 4.** Significant association between case/control status and the presence of kmers mapping to PVL was consistent across early and late subsets.

DOI: https://doi.org/10.7554/eLife.42486.009

coding sequences (CDS) were highly significantly associated with disease, being present in 98/101 (97%) pyomyositis cases and 84/417 (20%) carriage controls (unadjusted OR 129.5, p=$10^{-17.9}$). Kmers tagging variation in the 389 bp region immediately upstream of the CDS were also strongly associated with disease (p=$10^{-21.4}$). The most significant of these kmers were co-present with the CDS in the same cases (98/101, 97.0%), but present in fewer controls (79/417, 18.9%), producing an OR of 140.

Closer examination of this ~400 bp upstream region in genomes assembled from short-read Illumina sequencing showed that assembly of the region was problematic, with breaks or gaps in the assembly (*Figure 2—figure supplement 2*). To improve the accuracy of this region of the assembled genomes we performed long-read Oxford Nanopore sequencing on the 37 genomes with incomplete or discontinuous assembly upstream of the PVL CDS. By integrating long-read and short-read data we were able to assemble a single contig spanning this region in all isolates (*Figure 2—figure supplement 2*). When these improved assemblies were introduced, the signal of association upstream of the PVL CDS was no more significant than within the CDS (*Figure 2C*). Therefore, the presence of genomic sequence spanning the PVL toxin-coding sequences and the upstream, presumed regulatory, region exhibited the strongest association with pyomyositis in the *S. aureus* genome. All isolates with kmers mapping to the PVL CDS had 98% or more coverage for the PVL CDS genes in de novo assembly (*Figure 2—figure supplement 3*). The signal of association in the earlier and later periods of the study were examined and found to be consistent (*Figure 2—figure supplement 4*).

Out of 9175 kmers significantly associated with pyomyositis, we only found 101 kmers related to regions outside the PVL-carrying prophage (*Figure 2A*, *Supplementary file 2*). Two kmers mapping at a position near 0.2 Mb in the PYO2014 reference genome showed homology to platelet adhesin *sraP* by BLAST. Thirty-five kmers mapping to a 50 bp non-coding fragment at 0.6 Mb and two kmers mapping to 2.8 Mb showed homology to an MSSA476 intergenic sequence between adhesin-encoding *sdrC* and *sdrD* by BLAST. One kmer mapping to position 2.0 Mb showed no sequence homology by BLAST. Sixty-one kmers mapping to a 61 bp non-coding region at 2.7 Mb showed homology to an MSSA476 intergenic sequence between acetyltransferase-encoding genes SAS2453 and SAS2454 by BLAST. In conclusion, these other signals were short, fragmentary and mostly non-coding so we did not investigate them further.

The presence of high-risk kmers mapping to the PVL region explained the vast majority of observed heritability. When the presence or absence of the most significant kmer pattern, a set of kmers with an identical pattern of presence in the population, all of which mapped to the PVL region, was included as a covariate in GEMMA, the remaining heritability not explained by other factors was estimated and found to be 0.0% (95% CI 0–2.5%). Thus, the point estimate for heritability (not explained by the inclusion of PVL-tagging kmers) is reduced by 100% (95% CI 93.5–100%), meaning we have little evidence for any remaining heritability in case/control status.

Presence or absence of the PVL region accounted for the differences in pyomyositis rates between lineages. It was common in pyomyositis-associated lineages including CC-121 and absent from carriage/non-pyomyositis-associated lineages including ST-834 and CC-45 (*Figure 1*), explaining over 99.9% of observed heritability in case-control status. It was infrequent in the non-pyomyositis-associated sub-lineage of CC-121 (2/36, 5.6%), and sporadically present in pyomyositis cases in otherwise non-pyomyositis-associated, PVL-negative strains CC-1 and CC-88. Its absence from only three cases (in lineages CC-88, CC-1 and CC-121) suggested that the PVL region approached necessity for development of pyomyositis in the current setting in Cambodian children, while its presence in 20% of controls indicated that PVL-associated pyomyositis is incompletely penetrant, that is presence of the PVL region does not always lead to disease.

PVL genes were carried on multiple genetic backgrounds in this population, and the phage backgrounds vary by clonal complex. We examined all assemblies for sequence similarity to six known bacteriophages that carry the PVL genes, (*Boakes et al., 2011*) as well as the 45.7 kb region in PYO2134, a hypothesised integrated prophage identified in the reference genome prepared for this study, which we have called φCC121 (*Figure 2—figure supplement 3*). The finding of PVL genes on BLAST corresponded completely with the presence of kmers mapping across the PVL locus. We find sequences with >95% homology to four of these seven phages in the population. Regions in some assemblies showed homology for multiple phages, reflecting the similarity between φSLT, φSa2USA and φCC121 rather than the presence of multiple phages, and resolution of phages was limited by

fragmented assemblies from short reads (Supplementary File 2B). Phage types were restricted within most lineages, with jSLT found in ST-3206, jSa2USA in ST-1232 and jPVL in CC-1. jCC121 was the phage most often identified in the dominant pyomyositis strain CC-121, but it was absent in all but one isolate from the low risk subclade within CC-121. Strikingly, PVL-negative isolates in CC-121 and ST-834 strains frequently retained sequence homologous to >95% of jSLT, suggesting that some PVL-negative CC-121 isolates lost PVL secondarily by gene deletion rather than prophage excision.

## Discussion

In this study, we found a strong association between pyomyositis, a highly distinctive tropical infection of skeletal muscle in children, and Panton-Valentine leukocidin, a bacterial toxin commonly carried by bacteriophages. We found that a single coding region together with the upstream sequence are all but necessary for the development of pyomyositis: its sporadic presence is associated with pyomyositis in otherwise low-frequency strains, and its absence is associated with asymptomatic carriage in a high-propensity strain. PVL appears to be carried on multiple phage backgrounds in this population, but PVL-positive lineages generally carry a single phage type, as expected given the observed strain restriction of phages in *S. aureus* (*Xia and Wolz, 2014*; *Stegger et al., 2014*). The locally common PVL-positive CC-121 lineage contributes most strongly to the prevalence of pyomyositis in Cambodian children.

While PVL has long been thought an important *S. aureus* virulence factor, (*Diep et al., 2008*; *Bocchini et al., 2006*; *Kurt et al., 2013*) its role in invasive disease has been controversial, (*Otto, 2011*; *Day, 2013*) with conflicting results in case-control studies and an absence of supporting evidence on meta-analysis (*Shallcross et al., 2013*). In previous studies, the PVL-positive USA300 lineage was associated with musculoskeletal infection (both pyomyositis and osteomyelitis), however in these studies almost all such infections were caused by the USA300 strain, so the role of PVL was almost completely confounded by both methicillin resistance and strain background (*Pannaraj et al., 2006*; *Bocchini et al., 2006*). In our study, this confounding influence is broken down by the movement of PVL on mobile genetic elements (MGEs). Despite the emergence of CA-MRSA in carriage in Cambodia (*Nickerson et al., 2011*), all the pyomyositis cases were MSSA (*Figure 1—figure supplement 1*). By applying a GWAS method to a well-powered cohort, our study resolves the controversy around pyomyositis and PVL, demonstrating strong heritability which localises to a single region, even when the full bacterial genome is considered. Bacterial GWAS can pinpoint virulence variants when MGEs act to unravel linkage disequilibrium, if effect sizes are sufficiently strong.

There is strong biological plausibility for the association demonstrated in this study. PVL is a well characterised *S. aureus* toxin, toxic to the myeloid cells that form a first line of defence against bacterial infection, (*Oliveira et al., 2018*) and, in binding to myeloid cells by a complement receptor (C5aR), exerts immunomodulatory effects (*Spaan et al., 2013*). The establishment of muscle abscesses is a critical step in the pathogenesis of pyomyositis, but unlike renal, hepatic and splenic abscesses, skeletal muscle abscesses are rarely observed in experimental models of bacteraemia, unless there is preceding muscle trauma (*Miyake, 1904*). *S. aureus* strains containing PVL show increased duration of bacteraemia in a rabbit model of sepsis, (*Xia and Wolz, 2014*) and result in larger muscle abscesses (*Tseng et al., 2009*). PVL has been found strongly bound to necrotic muscle in an individual with myositis associated with necrotizing fasciitis (*Lehman et al., 2010*). These observations support the hypothesis that PVL may facilitate bacterial seeding to muscles via the bloodstream and tropism for muscular infection.

These results establish that, for children in Cambodia, staphylococcal pyomyositis is a disease whose pathogenesis depends crucially on a single toxin. This property is shared by toxin-driven, vaccine-preventable diseases such as tetanus and diphtheria. Therefore, vaccines that generate neutralising anti-toxin antibodies against PVL (*Landrum et al., 2017*) might protect human populations specifically against this common tropical disease. These results also raise the hypothesis that antibiotics which decrease toxin expression, and have been recommended in some PVL-associated infections, (*Saeed et al., 2018*) may offer specific clinical benefit in treating pyomyositis. More generally, our study provides an example of how microbial GWAS can be used to elucidate the pathogenesis of bacterial infections and identify specific virulence genes associated with human disease.

## Materials and methods

### Patient sample collection

We retrospectively identified pyomyositis cases from the Angkor Hospital for Children in Siem Reap, Cambodia, between January 2007 and November 2011. We screened all attendances in children (≤16 years) using clinical coding (ICD-10 code M60 (myositis)) and isolation of *S. aureus* from skeletal muscle abscess pus. We reviewed clinical notes to confirm a clinical diagnosis of pyomyositis was made by the medical staff, and bacterial strains cultured by routine clinical microbiology laboratory were retrieved from the local microbiology biobank. 106 clinical episodes of pyomyositis were identified, in 101 individuals, and we included the earliest episode from each individual.

We identified *S. aureus* nasal colonisation from two cohort studies undertaken at Angkor Hospital for Children. The first were selected from a collection characterising nasal colonization in the region between September and October, 2008, which has previously been described using multi-locus sequence typing (*Nickerson et al., 2011*). The swabs had been saved at −80°C since the study, these samples were re-examined for the presence of *S. aureus* using selective agar, confirmed using Staphaurex (Remel, Lenexa, USA) and the DNAse agar test (Oxoid, Hampshire, UK). Antimicrobial susceptibility testing was performed according to the 2014 Clinical and Laboratory Standards Institute guidelines (M100-24) (*CLSI, 2014*).

We undertook a second cohort study in 2012. Inclusion criteria were children (≤16 years) attending as an outpatient at Angkor Hospital for Children with informed consent. There were no exclusion criteria. Children were swabbed between the 2-7th July 2012, using sterile cotton tipped swabs pre-moistened (with phosphate buffered saline, PBS) using three full rotations of the swab within the anterior portion of each nostril with one swab being used for both nostrils, the ends were broken into bottles containing sterile PBS and kept cool until plated in the laboratory (within the hour). The swabs were plated onto Mannitol Salt agar to select for *S. aureus*. The M100-24 CLSI (*CLSI, 2014*) standards were followed for susceptibility testing and bacteria stored in tryptone soya broth and glycerol at −80°C.

We selected controls from carriers in these two cohorts using the excel randomization function: 222 of 519 from the 2008 cohort and 195 of 261 from the 2012 cohort.

### Ethical framework

Approval for this study was provided by the AHC institutional review board and the Oxford Tropical Ethics Committee (507-12).

### Whole genome sequencing

For each bacterial culture, a single colony was sub-cultured and DNA was extracted from the sub-cultured plate using a mechanical lysis step (FastPrep; MPBiomedicals, Santa Ana, CA) followed by a commercial kit (QuickGene; Autogen Inc, Holliston, MA), and sequenced at the Wellcome Centre for Human Genetics, Oxford on the Illumina (San Diego, California, USA) HiSeq 2500 platform, with paired-end reads 150 base pairs long.

A subset of samples were sequenced using long-read sequencing technology. We selected 37 isolates with incomplete assembly upstream of the PVL locus, 22 with ambiguous base calls in the assembly, and 15 where the region was assembled over two contigs. DNA was extracted using Genomic Tip 100/G (Qiagen, Manchester, UK) and DNA libraries prepared using Oxford Nanopore Technologies (ONT) SQK-LSK108 library kit (ONT, Oxford, UK) according to manufacturer instructions. These were then sequenced on ONT GridION device integrated with a FLO-MIN106 flow cell (ONT, Oxford, UK). ONT base calling was performed using Guppy v.1.6.

### Variant calling

For short-read sequencing, we used Velvet (*Zerbino and Birney, 2008*) v1.0.18 to assemble reads into contigs de novo. Velvet Optimiser v2.1.7 was used to choose the kmer lengths on a per sequence basis. The median kmer length was 123 bp (IQR 119–123). To obtain multilocus sequence types we used BLAST (*Altschul et al., 1990*) to find the relevant loci, and looked up the nucleotide sequences in the online database at http://saureus.mlst.net/. Strains that shared 6 of 7 MLST loci were considered to be in the same Clonal Complex (*Feil et al., 2003*). Antibiotic sensitivity was

predicted by interrogating the assemblies for a panel of resistance determinants as previously described (*Gordon et al., 2014*).

We used Stampy (*Lunter and Goodson, 2011*) v1.0.22 to map reads against reference genomes (USA300_FPR3757, Genbank accession number CP000255.1) (*Diep et al., 2006*). Repetitive regions, defined by BLAST (*Altschul et al., 1990*) comparison of the reference genome against itself, were masked prior to variant calling. Bases were called at each position using previously described quality filters (*Didelot et al., 2012*; *Young et al., 2012*; *Golubchik et al., 2013*).

After filtering ONT reads with filtlong v.0.2.0 (with settings filtlong – min_length 1000 –keep_percent 90 –target_bases 500000000 –trim –split 500), hybrid assembly of short (Illumina) and long (ONT) reads were made, using Unicycler v0.4.5 (*Wick et al., 2017*) (default settings). The workflow for these assemblies is available at https://gitlab.com/ModernisingMedicalMicrobiology/MOHAWK)

## Reconstructing the phylogenetic tree

We constructed a maximum likelihood phylogeny of mapped genomes for visualization using RAxML (*Stamatakis, 2014*) assuming a general time reversible (GTR) model. To overcome a limitation in the presence of divergent sequences whereby RAxML fixes a minimum branch length that may be longer than a single substitution event, we fine-tuned the estimates of branch lengths using ClonalFrameML (*Didelot and Wilson, 2015*).

## Kmer counting

We used a kmer-based approach to capture non-SNP variation (*Sheppard et al., 2013*). Using the de novo assembled genome, all unique 31 base haplotypes were counted using dsk (*Rizk et al., 2013*). If a kmer was found in the assembly, it was counted present for that genome, otherwise it was treated as absent. This produced a set of 10,744,013 variably present kmers, with the presence or absence of each determined per isolate. We identified a median of 2,801,000 kmers per isolate, including variably present kmers and kmers common to all genomes (IQR 2,778,000–2,837,000).

## Calculating heritability

We used the Genome-wide Efficient Mixed Model Association tool (GEMMA (*Zhou and Stephens, 2012*)) to fit a univariate linear mixed model for association between a single phenotype (pyomyositis vs asymptomatic nasal carriage). We calculated the relatedness matrix from kmers, and used GEMMA to estimate the proportion of variance in phenotypes explained by genotypic diversity in the sample set (i.e. estimated heritability). Heritability estimates with and without a covariate (e.g. the presence of high risk kmers) are compared by testing for difference in proportions. We use the point estimate for heritability as the denominator to calculate the relative decrease proportion.

## Genome wide association testing of kmers

We performed association testing using an R package bacterialGWAS (https://github.com/jessiewu/bacterialGWAS), which implements a published method for locus testing in bacterial GWAS (*Earle et al., 2016*). The association of each kmer on the phenotype was tested, controlling for the population structure and genetic background using a linear mixed model (LMM) implemented in GEMMA (*García et al., 2013*). The parameters of the linear mixed model were estimated by maximum likelihood and likelihood ratio test was performed against the null hypothesis (that each locus has no effect) using the software GEMMA (*García et al., 2013*). GEMMA was run using a minor allele frequency of 0 to include all SNPs. GEMMA was modified to output the ML log-likelihood under the null, and alternative and $-\log_{10} p$ values were calculated using R scripts in the bacterialGWAS package. Unadjusted odds ratios were reported because there was no residual heritability unexplained by the most significant kmers.

To address the possibility of differing effect sizes between the two control cohorts, we have repeated the analysis after splitting the study into two groups – early (2008 and earlier, n = 276, cases n = 54, controls n = 222) and late (2009 and later, n = 242, cases n = 47, controls n = 195). We then examined the maximum likelihood estimates produced by the LMM for kmers mapping to the PVL coding sequence in each region. The 95% CI of the estimate from each sub study were overlapping (See *Figure 2—figure supplement 4*).

### Testing for lineage effects

We tested for associations between lineage and phenotype using an R package *bugwas* (available at https://github.com/sgearle/bugwas), which implements a published method for lineage testing in bacterial GWAS (*Earle et al., 2016*). We tested lineages using principal components. These were computed based on biallelic SNPs using the R function prcomp. To test the null hypothesis of no background effect of each principal component, we used a Wald test, which we compared against a $\chi$ (*Chewapreecha et al., 2014*) distribution with one degree of freedom to obtain a *p* value.

### Kmer mapping

We used Bowtie (*Langmead and Salzberg, 2012*) to align all 31 bp kmers from short-read sequencing to a draft reference (the de novo assembly of a CC-121 pyomyosits strain PYO2134). Areas of homology between the draft reference and well-annotated reference strains were identified by aligning sequences with Mauve (*Darling et al., 2004*). For all 31 bp kmers with significant association with case-controls status, the likely origin of the kmer was determined by nucleotide sequence BLAST (*Altschul et al., 1990*) of the kmers against a database of all *S. aureus* sequences in Genbank.

### Joint short-read and long-read analysis

31 bp kmers were counted for the 37 hybrid short-read and long-read assemblies using dsk (*Rizk et al., 2013*). The presence or absence of all Illumina (short-read) kmers that mapped to the two PVL toxin-coding sequences and the upstream intergenic region plus the surrounding 1 kb were reassessed. For the 37 samples with hybrid assemblies, the presence/absence of these kmers was determined from the kmers counted from the hybrid assemblies. For all other samples, presence/absence was determined from the kmers counted from the short-read only assemblies. The new presence/absence patterns were tested for association with the phenotype controlling for population structure and genetic background using GEMMA (*Zhou and Stephens, 2012*), using the same relatedness matrix as the original short-read analysis.

### Predicting presence of PVL genes and bacteriophages

We used BLAST to check for the relative coverage of the PVL CDS (From reference genome USA300_FPR3757 (CP000255.1) positions 1546170–1548350), as well as the entire sequence of 6 known PVL positive phages (φ2958(NC_011344.1), φPVL (NC_002321.1), φPVL108 (NC_008689.1), φSLT (NC_002661.2), φSa2MW (NC_003923.1), φSa2USA (CP000255.1)) (*Boakes et al., 2011*), as well as the hypothesised prophage region from PYO2134 (1571177–1616957), which we have calledφCC121. For PVL genes, we determined relative coverage of the query sequence; over 98% coverage was used as threshold for gene presence.

### Multiple testing correction

Multiple testing was accounted for by applying a Bonferroni correction (*Dunn, 1959*); the individual locus effect of a variant (kmer or PC) was considered significant if its *P* value was smaller than $\alpha/n_p$, where we took $\alpha = 0.05$ to be the genome-wide false-positive rate and $n_p$ to be the number of kmers or PCs with unique phylogenetic patterns, that is, unique partitions of individuals according to allele membership. We identified 236627 unique kmer patterns and 518 PCs, giving thresholds of $2.1 \times 10^{-7}$ and $9.7 \times 10^{-5}$ respectively.

### Data availability

Sequence data have been submitted to Short Read Archive (Bioproject ID PRJNA418899). Clinical origins of sequenced strains are listed in supplementary information (*Supplementary File 4*).

## Acknowledgments

The authors would like to thank study participants. This study was funded by the Wellcome Trust (MORU Grants 089275/H/09/Z and 089275/Z/09/Z), and a University of Oxford Medical Research Fund awarded to CEM (MRF/MT2015/2180). DJW is a Sir Henry Dale Fellow, jointly funded by the Wellcome Trust and the Royal Society (Grant 101237/Z/13/Z). BCY is a Research Training Fellow

funded by the Wellcome Trust (Grant 101611/Z/13/Z). DHW was funded by the National Institute for Health Research (NIHR) Oxford Biomedical Research Centre (BRC) and the European Union's Seventh Framework Programme under the grant agreement number 601783 (BELLEROPHON project). NS is funded by a Public Health England (PHE)/University of Oxford Clinical Lectureship. KE was funded by an academic clinical fellowship which was provided by the UK NIHR through the University of Oxford. This research was supported by Core funding to the Wellcome Centre for Human Genetics provided by the Wellcome (090532/Z/09/Z). The views expressed are those of the author(s) and not necessarily those of the NHS, PHE, the NIHR or the Department of Health.

## Additional information

### Funding

| Funder | Grant reference number | Author |
| --- | --- | --- |
| Wellcome | 090532/Z/09/Z | Rory Bowden |
| Seventh Framework Programme | 601783 | David H Wyllie |
| Wellcome | 089275/H/09/Z | Nicholas PJ Day |
| Wellcome | 089275/Z/09/Z | Nicholas PJ Day |
| Royal Society | 101237/Z/13/Z | Daniel J Wilson |
| National Institute for Health Research | | Daniel J Wilson |
| University Of Oxford | MRF/MT2015/2180 | Catrin E Moore |

The funders had no role in study design, data collection and interpretation, or the decision to submit the work for publication.

### Author contributions

Bernadette C Young, Investigation, Visualization, Methodology, Writing—original draft, Writing—review and editing; Sarah G Earle, Data curation, Formal analysis, Methodology, Writing—review and editing; Sona Soeng, Poda Sar, Investigation, Methodology; Varun Kumar, Conceptualization, Writing—review and editing; Songly Hor, Vuthy Sar, Nicole Stoesser, Paul Turner, Rory Bowden, Derrick W Crook, David H Wyllie, Investigation, Writing—review and editing; Rachel Bousfield, Data curation, Investigation, Writing—review and editing; Nicholas D Sanderson, Leanne Barker, Emma K Nickerson, Investigation, Methodology, Writing—review and editing; Katherine RW Emary, Conceptualization, Investigation, Writing—review and editing; Christopher M Parry, Conceptualization, Investigation, Methodology, Writing—review and editing; Nicholas PJ Day, Conceptualization, Supervision, Writing—review and editing; Daniel J Wilson, Resources, Supervision, Writing—original draft, Writing—review and editing; Catrin E Moore, Conceptualization, Resources, Data curation, Formal analysis, Supervision, Funding acquisition, Investigation, Methodology, Writing—original draft, Project administration, Writing—review and editing

### Author ORCIDs

Bernadette C Young (iD) http://orcid.org/0000-0001-6071-6770
Sarah G Earle (iD) http://orcid.org/0000-0002-0738-7557
Christopher M Parry (iD) http://orcid.org/0000-0001-7563-7282
Paul Turner (iD) http://orcid.org/0000-0002-1013-7815
Rory Bowden (iD) http://orcid.org/0000-0001-8596-0366
Derrick W Crook (iD) http://orcid.org/0000-0002-0590-2850
Daniel J Wilson (iD) http://orcid.org/0000-0002-0940-3311
Catrin E Moore (iD) http://orcid.org/0000-0002-8639-9846

### Ethics

Human subjects: Approval for this study was provided by the AHC institutional review board and the Oxford Tropical Ethics Committee (507-12).

### Decision letter and Author response

Decision letter https://doi.org/10.7554/eLife.42486.018
Author response https://doi.org/10.7554/eLife.42486.019

## Additional files

### Supplementary files

• Supplementary file 1. Isolates included in this study.
DOI: https://doi.org/10.7554/eLife.42486.010

• Supplementary file 2. All significant kmers from short-read sequencing assembly, evidence of association, frequency, location on mapping to the study reference PYO2134, best match on BLAST (blastn) to all *S. aureus* coding sequences in Genbank, and best match on BLAST (blastn) to the NCBI database.
DOI: https://doi.org/10.7554/eLife.42486.011

• Supplementary file 3. Presence of high risk kmers and relative coverage of PVL coding sequence and common PVL positive phages found by BLAST (blastn) of short read assemblies.
DOI: https://doi.org/10.7554/eLife.42486.012

• Supplementary file 4. List of all isolates, and site of isolation (carriage or invasive disease) and year of isolation. These isolate names match those used in sequence data deposition on SRA.
DOI: https://doi.org/10.7554/eLife.42486.013

• Transparent reporting form
DOI: https://doi.org/10.7554/eLife.42486.014

### Data availability

Sequence data has been submitted to Short Read Archive (Bioproject ID PRJNA418899).

The following dataset was generated:

| Author(s) | Year | Dataset title | Dataset URL | Database and Identifier |
|---|---|---|---|---|
| Young BC, Earle SG, Soeng S, Sar P, Kumar V, Hor S, Sar V, Bousfield R, Sanderson ND, Barker L, Stoesser N, Emary KRW, Parry CM, Nickerson EK, Turner P, Bowden R, Crook D, Wyllie D, Day NPJ | 2018 | A genome-wide association study of S. aureus cultured from 101 children with pyomyositis and 417 children with asymptomatic nasal carriage attending the Angkor Hospital for Children in Cambodia | https://www.ncbi.nlm.nih.gov/bioproject/?term=PRJNA418899 | NCBI Bioproject, PRJNA418899 |

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
