## [Decision Letter]

Thank you for submitting your article "Panton-Valentine leukocidin is the key determinant of *Staphylococcus aureus* pyomyositis in a bacterial GWAS" for consideration by *eLife*. Your article has been reviewed by three peer reviewers, one of whom is a member of our Board of Reviewing Editors, and the evaluation has been overseen by Wendy Garrett as the Senior Editor. The following individual involved in the review of your submission has agreed to reveal his identity: John Lees (Reviewer #2).

The reviewers have discussed the reviews with one another and the Reviewing Editor has drafted this decision to help you prepare a revised submission.

Summary:

The authors use genome-wide association approaches to identify Panton-Valentine leucocidin as the primary determinant of disease causation in *S. aureus* pyomyositis.

Pyomyositis is purulent infection of skeletal muscles and associated with abscess formation. There is a strong association of the pathogen *S. aureus* with pyomyositis, 90% of cases in tropical regions, and ~75% in temperate regions. This study, by a team with strong track record in tropical medicine, and GWAS, links Panton-Valentine Leucocidin (PVL), a prophage-associated virulence determinant that is variably found in the *S. aureus* population, to pyomyositis. Their work provides strong evidence for the important role of the PVL toxin in pyomyositis in a paediatric cohort in Cambodia. Whilst there have been some controversies about the role that PVL plays in particular models of disease, there is strong clinical evidence for the association of PVL with some diseases, particularly skin and soft tissue infections (SSTIs), where the formation of boils and carbuncles are common. Abscess formation is a common link between pyomyositis and SSTIs.

The manuscript is a strong contribution to the field of association studies in bacteria. They have used a carefully collected, well-matched and well phenotyped sample collection, and properly applied cutting edge mathematical approaches. The design of the study, both in the sampling and use GWAS for analysis, is appropriate for answering the question the authors set out. Also valuable is the use of long-read sequencing to resolve the region of interest in all samples. The manuscript will settle some of the controversy about the role of PV toxin, at least in this disease context.

This paper has the potential to bridge the gap from candidate gene studies of PVL to more useful GWAS approaches, possibly leading to further meta-analysis in other populations.

Essential revisions:

1) In the Introduction and Discussion the authors state that they are trying to expand GWAS from the identification of variants under very strong selection, such as antibiotic resistance, into more subtle phenotypes with lower effects. However, there are two issues with this: i) bacterial GWAS has already been used to identify variants underlying subtle and non-resistance phenotypes. For just a few examples, see their reference 1, which analysed host preference in Campylobacter, their reference 4, which identified variants associated with invasiveness in S. pyogenes, PMID: 28742023, which looked at determinants of carriage duration in S. pneumoniae, PMID: 27997216, PMID: 28112723, PMID: 28785103, to name a few. The point about the strong selection on AMR vs. MGEs breaking up population structure is useful to make here, but instead of setting up an implication of this being the first non-proof-of-principle GWAS, please cite these studies instead. ii) The variant they have found has an enormous odds ratio, and is clearly under very strong selection. These sections should be toned down and the relevant literature cited.

2) It is unclear from the manuscript how valid the question the authors set out to answer actually is i.e. how controversial is the role of PVL in SSTIs? Reading the introduction and some of the cited references it does look like epidemiological studies have generally supported a role of PVL in SSTIs. Taken together with work in animal models, which seem to have been overlooked in this paper's introduction and references (an example of such a reference would be PMID 23684309), one might view this study as a confirmation of the role of PVL, rather than resolving an ongoing debate. The Introduction therefore needs more details and referencing to describe the controversy of the PVL association, whether this is really an ongoing debate, and why GWAS is needed to resolve this.

3) It was not clear how the high proportion of heritability explained by the PVL hit was estimated. This is a major result of this paper and should be clear. Furthermore, given the small number of samples, is it possible to provide a confidence interval, perhaps based on bootstrapping?

4) The use of non-USA/MRSA samples is an important strength of this study (as the authors note). Can the authors provide a more quantitative justification for this advantage? It is not necessary to perform a full meta-analysis or replication study in another population, but a summary of PVL vs SSTI presence/absence in another (more frequently studied e.g. MRSA) population would be useful to support this argument.

5) Given the multifaceted mechanisms of *S. aureus* that modulate host-cell interactions, it is not unreasonable to suppose that there are multiple factors that contribute to pathogenic processes involved in pyomyositis. From a pathological point of view, prior to the formation of abscesses that characterize pyomyositis, there is haematogenous spread of the pathogen, and therefore survival in the blood will be an important step in the disease process. In the authors' analysis PVL clearly demonstrates a strong association. However, the paper does not extend this to examine if there are other signals beyond PVL. Figure 2A shows that there are other associations outside the prophage reaching significance and which may provide clues to other determinants contributing to disease. These should be reported and discussed.

6) The authors have generated a strong hypothesis for the role of PVL in disease, but this is where the study ends. There is no mechanistic support for this through experimentation, or any discussion, which relates it to the pathology of disease or the cellular mechanisms involved. We agree that it is beyond the scope of the study to perform experimental work to probe the mechanistic link, but it would be helpful if the discussion were to explore the mechanistic basis of their observations and link it to pathogenicity.

7) The authors do not explore the relationship of the PVL carrying prophages within the *S. aureus* population in the study. This is an important aspect of the genetics of *S. aureus* of this data set that impinges on disease potential. What is the diversity of the prophages observed, where are they integrated, is there evidence of PVL mobility within and between lineages? This is an element of analysis that would add to the story, and link this back to the pathogen and our understanding of its ability to cause disease.

---

## [Author Response]

Essential revisions:1) In the Introduction and Discussion the authors state that they are trying to expand GWAS from the identification of variants under very strong selection, such as antibiotic resistance, into more subtle phenotypes with lower effects. However, there are two issues with this: i) bacterial GWAS has already been used to identify variants underlying subtle and non-resistance phenotypes. For just a few examples, see their reference 1, which analysed host preference in Campylobacter, their reference 4, which identified variants associated with invasiveness in S. pyogenes, PMID: 28742023, which looked at determinants of carriage duration in S. pneumoniae, PMID: 27997216, PMID: 28112723, PMID: 28785103, to name a few. The point about the strong selection on AMR vs. MGEs breaking up population structure is useful to make here, but instead of setting up an implication of this being the first non-proof-of-principle GWAS, please cite these studies instead. ii) The variant they have found has an enormous odds ratio, and is clearly under very strong selection. These sections should be toned down and the relevant literature cited.

We would like to thank the reviewers for these important points. We have modified our Introduction to include recent results, and note the phenotypes where GWAS has demonstrated an association (host specificity, invasive disease with *Streptococcus pyogenes* and *Staphylococcus epidermidis*, pneumococcal carriage, mortality following bacteraemia with CC-30 *Staphylococcus aureus*, endemic melioidosis). We have noted in discussion that the strong effect size contributes the success of this study. We have moderated the language in discussion about the importance of MGEs by noting that a large effect size was present in this study.

2) It is unclear from the manuscript how valid the question the authors set out to answer actually is i.e. how controversial is the role of PVL in SSTIs? Reading the introduction and some of the cited references it does look like epidemiological studies have generally supported a role of PVL in SSTIs. Taken together with work in animal models, which seem to have been overlooked in this paper's introduction and references (an example of such a reference would be PMID 23684309), one might view this study as a confirmation of the role of PVL, rather than resolving an ongoing debate. The Introduction therefore needs more details and referencing to describe the controversy of the PVL association, whether this is really an ongoing debate, and why GWAS is needed to resolve this.

It is important to clarify that our study is of pyomyositis, not skin and soft tissue infections. As the reviewers observe, PVL is widely accepted as a virulence factor for superficial skin and soft tissue infections such as abscesses, boils and furunculosis (see Shallcross et al., 2013 and Saeed et al., 2017). However pyomyositis remains a distinct entity of invasive disease observed in tropical areas, with the formation of purulent infection within skeletal muscle, while the overlying skin remains intact. Unlike infections of the skin and superficial soft tissues, the skin and subcutaneous tissues are not usually involved in pyomyositis, by contrast to intense inflammation in the infected muscles. Pyomyositis is thought to arise from haematogenous seeding of bacteria from blood to muscle (See Verma et al., 2016). Reviews of the role of PVL in disease – including the two cited here – generally separate it from skin and soft tissue infections, categorising it among musculoskeletal infections.

We have researched the question extensively and believe that the studies supporting and refuting the association of PVL are yet to provide an unequivocal answer to the question. In conducting this study, we started without a hypothesis that we would find specific causative genes associated with pyomyositis; our aim was to investigate whether any bacterial genetic factors contribute to the characteristic disease entity of pyomyositis. Given the striking results localised to a single gene, and because prominent results reporting association between PVL with other conditions (e.g. necrotising pneumonia) have been controversial, we chose to contextualise the results early with a discussion of PVL as a virulence factor, including what is known about its possible contribution to pyomyositis. However, the discussion of PVL in the introduction should not indicate that this was a study designed to investigate the role of PVL in pyomyositis. We have clarified this further in the Introduction.

We would strongly contend that no strong evidence establishes an association between PVL and pyomyositis prior to this study. Indeed, in the article on mechanisms of action of PVL suggested here (PMID 23684309), the authors note “the role of PVL is still under debate”. In the meta-analysis conducted by Shallcross et al., the authors find that in contrast to the strong association with skin and soft tissue infections, PVL strains are comparatively rare in musculoskeletal infections (OR 0.44, CI 0.19-0.99). In the ISAC position statement on PVL, they find the association between PVL and SSTI is highest for furuncuolisis. This review presents limited statements on musculoskeletal infection. They do note “PVL-SA are also associated with severe musculoskeletal infections, particularly in children”, but the only evidence they present (Sheikh et al., 2015, British Journal of hospital medicine) is a review specifically of osteomyelitis. It is notable that this review cites as evidence the meta-analysis by Shallcross et al., which found existing evidence for an association between musculoskeletal infection and PVL was weak. The review by Sheikh et al. further supports the claims of association with studies that report non-significant results on tests of association: “Panton-Valentine leukocidin-producing *S. aureus* osteomyelitis can be multifocal with multiple bones involved. Patients are also more likely to present with large sub-periosteal abscesses (risk ratio 3.90, *P*=0.25) […] and associated myositis and/or pyomyositis (risk ratio 3.22, *P*=0.18)[…].”

To address this concern, and to avoid any unintended confusion between pyomyositis and superficial SSTI, we have therefore included further detail throughout our introduction to reiterate the aims of this study (to undertake an unbiased survey of the whole genome, rather than hypothesis driven investigation), and its focus on the invasive, distinctive pathological entity of pyomyositis rather than superficial skin and soft tissue infection. We have also expanded the literature review to clearly state why we believe our results illuminate an ongoing debate rather than confirm a settled answer.

3) It was not clear how the high proportion of heritability explained by the PVL hit was estimated. This is a major result of this paper and should be clear. Furthermore, given the small number of samples, is it possible to provide a confidence interval, perhaps based on bootstrapping?

We determined this figure by including the presence of the most significant kmer, which tags PVL, as a covariate in the linear mixed model. When this is included, the estimate of heritability output by the model (in GEMMA), which represents the estimated proportion of heritability not explained by other factors, drops from 63.8% to 0.0%. Thus, the proportion of difference explained ((p1-p2)/p1) is over 99.99% (95% CI 93.5-100%). We have included this more detailed description of the heritability results (Results paragraph seven), including the confidence interval for proportion accounted for, and added to the relevant Materials and methods section (subsection “Genome wide association testing of Kmers.”).

4) The use of non-USA/MRSA samples is an important strength of this study (as the authors note). Can the authors provide a more quantitative justification for this advantage? It is not necessary to perform a full meta-analysis or replication study in another population, but a summary of PVL vs SSTI presence/absence in another (more frequently studied e.g. MRSA) population would be useful to support this argument.

With apologies, we do not understand what this question seeks to address. We do not believe that evidence about prevalence of PVL in skin and soft tissue infections can be directly applied to help our understanding of primary, invasive muscle infection. Regarding pyomyositis, limited studies exist. Our paper references a meta-analysis, which summarises the prevalence of PVL in existing high-quality studies of musculoskeletal infections. Only two of these studies were of pyomyositis (the rest being bone and joint infections), and neither had carriage control populations (one had skin and soft tissue infections as a comparator group, and the other was a case series without control group). The findings and limitations of studies of CA-MRSA and pyomyositis are detailed in the Discussion.

5) Given the multifaceted mechanisms of S. aureus that modulate host-cell interactions, it is not unreasonable to suppose that there are multiple factors that contribute to pathogenic processes involved in pyomyositis. From a pathological point of view, prior to the formation of abscesses that characterize pyomyositis, there is haematogenous spread of the pathogen, and therefore survival in the blood will be an important step in the disease process. In the authors' analysis PVL clearly demonstrates a strong association. However, the paper does not extend this to examine if there are other signals beyond PVL. Figure 2A shows that there are other associations outside the prophage reaching significance and which may provide clues to other determinants contributing to disease. These should be reported and discussed.

This is an important point, and the incomplete penetrance of PVL suggests other factors (host, bacterial or environmental) in the pathogenic process. However, we have little evidence for any remaining bacterial genetic heritability (see point 3 above for a formal demonstration of this). Following the reviewers’ suggestion, we have included a discussion of the analysis of significant kmers outside of the prophage region (Results paragraph six, Supplementary file 2). These findings are cautiously interpreted, given the very low remaining heritability once PVL is accounted for.

6) The authors have generated a strong hypothesis for the role of PVL in disease, but this is where the study ends. There is no mechanistic support for this through experimentation, or any discussion, which relates it to the pathology of disease or the cellular mechanisms involved. We agree that it is beyond the scope of the study to perform experimental work to probe the mechanistic link, but it would be helpful if the discussion were to explore the mechanistic basis of their observations and link it to pathogenicity.

We agree, this is a helpful addition to the observations of our study. We have included discussion on the extensive experimental work that has previously reported plausible mechanism for the action of PVL in muscle infections (Discussion section).

7) The authors do not explore the relationship of the PVL carrying prophages within the S. aureus population in the study. This is an important aspect of the genetics of S. aureus of this data set that impinges on disease potential. What is the diversity of the prophages observed, where are they integrated, is there evidence of PVL mobility within and between lineages? This is an element of analysis that would add to the story, and link this back to the pathogen and our understanding of its ability to cause disease.

We agree this is a helpful development of the study results. We have added a section in results discussing the diversity of phages within and between lineages (Discussion paragraph nine, (see also Figure 2—figure supplement 4, Supplementary file 3). There are challenges to the study of phages with short read sequencing data; in almost all isolates the likely phage sequences are fragmented, so an analysis of the integration of prophages is not currently possible. However we think the question of phage integration throughout the population, while an important question, is incidental to the current study’s main result.